# 4-Disubstituted Pyrazolin-3-Ones—Novel Class of Fungicides against Phytopathogenic Fungi

Elena R. Lopat'eva [1], Alexander S. Budnikov [1,2], Igor B. Krylov [1,2,3,*], Anna L. Alekseenko [1,3], Alexey I. Ilovaisky [1,2], Alexey P. Glinushkin [2] and Alexander O. Terent'ev [1,2,3,*]

1   N. D. Zelinsky Institute of Organic Chemistry, Russian Academy of Sciences, 47 Leninsky Prospekt, 119991 Moscow, Russia
2   All-Russian Research Institute for Phytopathology, B. Vyazyomy, 143050 Moscow Region, Russia
3   Faculty of Chemical and Pharmaceutical Technology and Biomedical Products, D. I. Mendeleev University of Chemical Technology of Russia, 9 Miusskaya Square, 125047 Moscow, Russia
*   Correspondence: krylovigor@yandex.ru (I.B.K.); alterex@yandex.ru (A.O.T.)

**Abstract:** The search for fungicides of novel classes is the long-standing priority in crop protection due to the continuous development of fungal resistance against currently used types of active compounds. Recently, 4-nitropyrazolin-3-ones were discovered as highly potent fungicides, of which activity was believed to be strongly associated with the presence of a nitro group in the pyrazolone ring. In this paper, a series of 4-substituted pyrazolin-3-ones were synthesized and their fungicidal activity against an important species of phytopathogenic fungi (*Venturia inaequalis*, *Rhizoctonia solani*, *Fusarium oxysporum*, *Fusarium moniliforme*, *Bipolaris sorokiniana*, and *Sclerotinia sclerotiorum*) was tested in vitro. We discovered that 4-mono and 4,4-dihalogenated pyrazolin-3-ones demonstrate fungicidal activity comparable to that of 4-nitropyrazolin-3-ones and other modern fungicides (such as kresoxim methyl). This discovery indicates that $NO_2$ moiety can be replaced by other groups of comparable size and electronic properties without the loss of fungicidal activity and significantly expands the scope of potent new fungicides based on a pyrazolin-3-one fragment.

**Keywords:** pyrazolin-3-ones; fungicidal compounds; crop protection; nitropyrazolones; new modes of action

## 1. Introduction

In today's world, the problem of microbial contamination of agricultural crops is critical to providing food for a growing human population [1]. Fungi represent one of the most harmful groups of phytopathogens, which account for up to 80% of crop losses [2–5]. Moreover, phytopathogenic fungi produce toxic metabolites, which represent a serious risk for public health as food contaminants [6–9]. Besides, fungal pathogens can cause opportunistic fungal infections in humans and animals [10,11].

Despite active research [12–14], the output of new fungicides has been relatively constant for the past 20 years [15]. To date, only a few classes of compounds dominate the market (77% of sales in 2018 [15]): Quinone outside Inhibitors (QoI, strobilurins, C3), De-Methylation Inhibitors (DMI, triazoles and imidazoles, G1) [16], Succinate-dehydrogenase inhibitors (SDHI, C2) [17], Dithiocarbamates (M03), Chloronitriles (M05), Carboxylic Acid Amides (CAA, H5) and Phenyl Amides (PA, A1), which has created the basis for the development of pest resistance [18]. In addition, the use of fungicides with the same mechanism of action both for agricultural needs and in medicine can create conditions for outbreaks of human diseases caused by ant. imycotic-resistant strains of fungi [19,20]. In this regard, the development of novel types of fungicides (First-in-Class compounds) is a necessary and urgent task [21].

The five-membered pyrazolin-3-one (pyrazolone) ring is a privileged structural motif in medicinal chemistry with a wide range of biological activities [22] (Figure 1). The

practical importance of pyrazolones attracts continuous interest to the development of their synthetic methodology and properties [23,24]. The history of the medical use of 4-unsubstituted pyrazolinones began as early as 1887 with the discovery of antipyrine (1,5-dimethyl-2-phenyl-1,2-dihydro-3H-pyrazol-3-one), one of the first nonopioid analgesics and antipyretics. This discovery prompted the study of pyrazolone derivatives, including C4-monosubstituted pyrazolones, pyrazolone-based Schiff bases, and their metal complexes, which possess anti-inflammatory, antipyretic and analgesic [22,25–27], antitumor/cytotoxic [22,25,28–30], antimicrobial [22,25,31–33], antioxidant [22,26], and protein denaturation inhibiting [27] activities. The interest of medicinal chemists to pyrazolones has been maintained and is increasing at the present time [22]. Recently, C4-disubstituted pyrazolones, primarily spiropyrazolones, have also attracted increasing attention as biologically active compounds [34–36]. Due to the wide spectrum of the biological activity of disubstituted pyrazolones, new asymmetric methods for their synthesis are being actively developed [36–41]. C4-disubstituted pyrazolones are recognized as valuable antitumor agents [35,42–46], antimicrobial substances [47], inhibitors of trypanosomal phosphodiesterase B1 [48], RalA inhibitors [49], and miticides [50]. However, their potential as effective fungicides has not been expected.

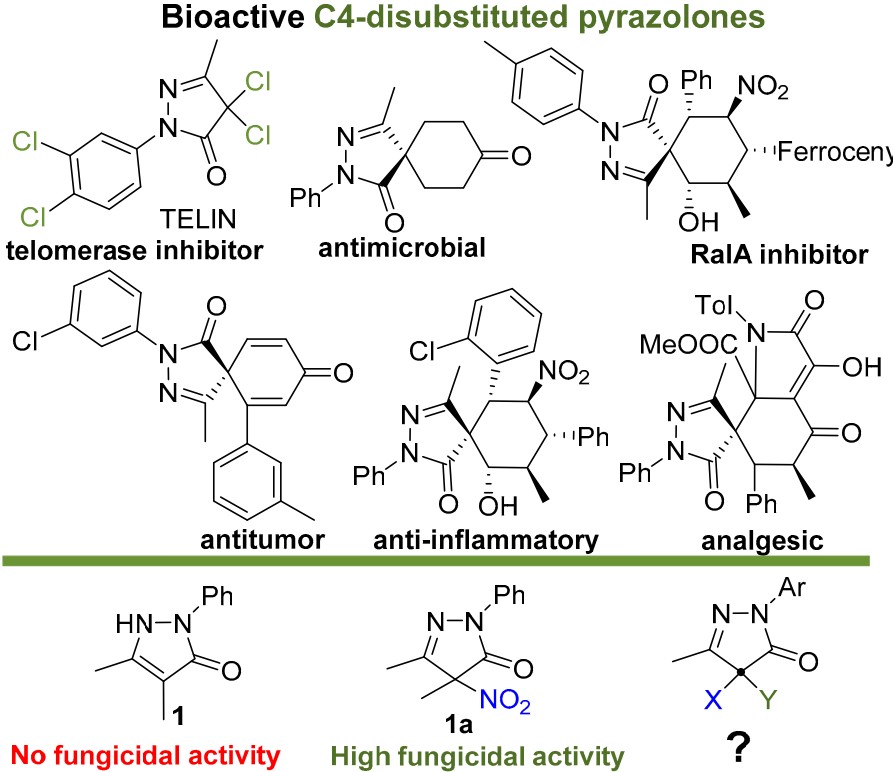

**Figure 1.** Selected examples of biologically active C4-disubtituted pyrazolin-3-ones.

In our previous reports, 4-nitropyrazoline-3-ones were discovered as the novel class of highly potent broad spectrum fungicides [51,52]. However, the understanding of the underlying modes of action was lacking. To shed light on this issue, it is desirable to know how the structure of pyrazolone derivatives affects their fungicidal activity. Our previous study [51] of the structure–activity relationship revealed the importance of the aromatic substituent at N2, a small alkyl substituents at the C4 and C5 atoms of the pyrazolone ring, and the C(sp³)-hybridized C4 atom for high fungicidal activity [51]. Since the unnitrated pyrazolone (**1**) did not exhibit significant activity [52] (Figure 1), fungicidal properties were believed to be associated with the presence of a nitro group in the pyrazolone ring. In the present work, we aimed to synthesize and study other pyrazolones with a C(sp³)-

hybridized C4 atom containing no nitro-group to reveal the role of substituents at this position for the fungicidal activity.

## 2. Results and Discussion

### 2.1. The Synthesis of the 4,4-Disubstituted Pyrazolone Derivatives

　　We have synthesized and tested a series of pyrazolones with variable substituent at position 4 instead of the $NO_2$ group to reveal the role of this substituent. The structure of one of the most active nitropyrazolones, 4,5-dimethyl-4-nitro-2-phenyl-2,4-dihydro-3H-pyrazol-3-one (**1a**, Scheme 1), was used as the reference [51,52].

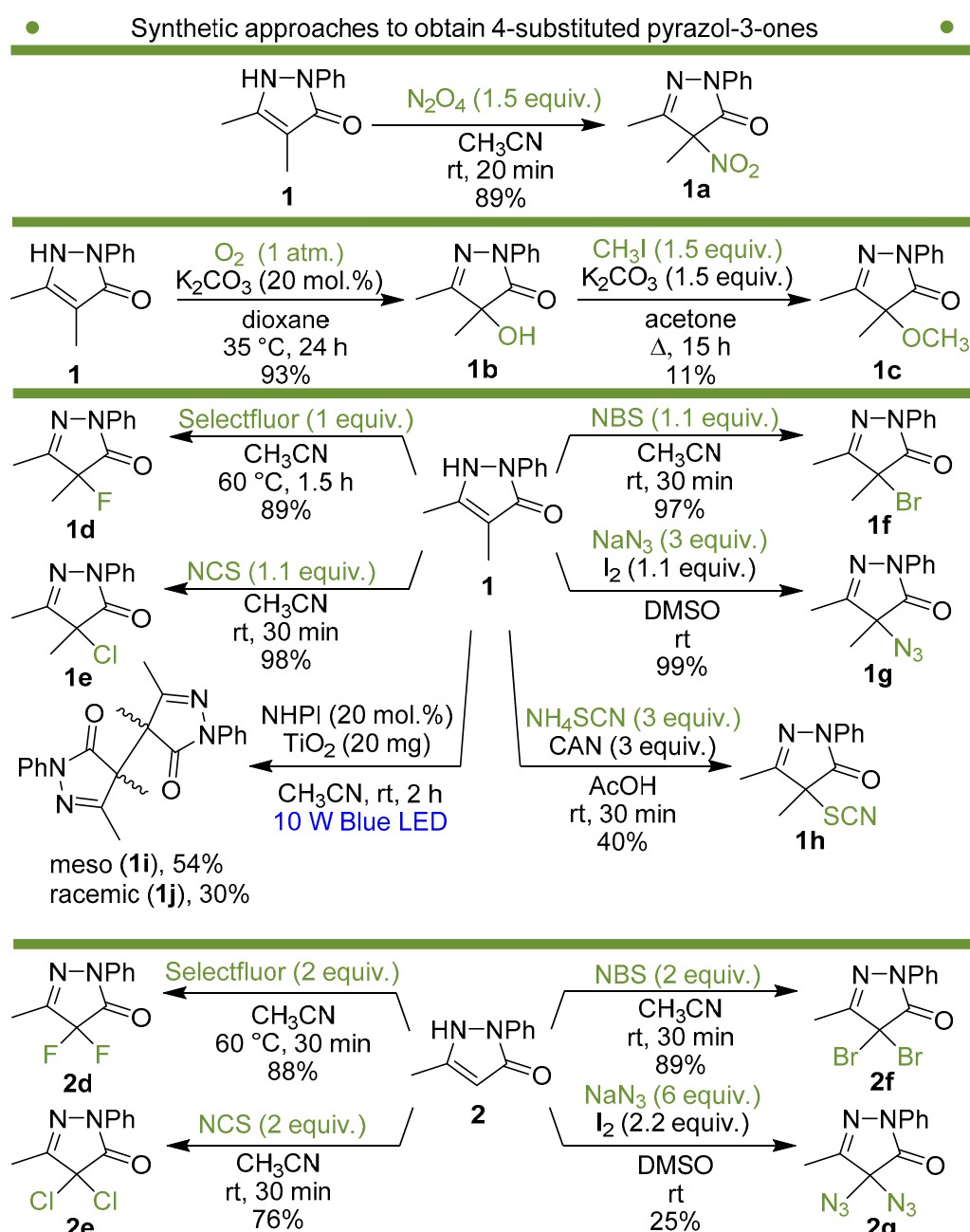

**Scheme 1.** Synthetic approaches to 4-substituted pyrazolin-3-ones used in the present study.

　　Synthetic approaches to the target compounds **1b–1j** and **2d–g** are shown in Scheme 1. A previously developed synthetic procedure using $N_2O_4$ as a nitrating agent [51] was used for the synthesis of **1a**. The advantages of this method are the absence of metal-containing

reagents in contrast to the metal salt/NaNO$_2$ system [52], high selectivity, and scalability up to multigram quantities without the yield drop and without the need for chromatographic purification of the target product. 4-Methyl-4-hydroxypyrazolone (**1b**) was synthesized by aerobic oxidation of 4-methyl-2-phenyl-5-dimethyl-pyrazolin-3-one (**1**) under basic conditions (K$_2$CO$_3$) [53]. The further methylation step gave 4-methoxypyrazolone (**1c**). Monohalogenated fluoro-, chloro- and bromopyrazolones (**1d–f**) were synthesized by the halogenation of pyrazolone **1** with Selectfluor™, N-chlorosuccinimide (NCS), and N-bromosuccinimide (NBS), respectively. It should be noted that these procedures provided up to quantitative yields. 4,4-Dihalogenated pyrazolones (**2d–f**) were synthesized similarly from 2-phenyl-5-dimethyl-pyrazol-3-one (**2**) employing two equivalents of halogenating agents. 4-Azido-4,5-dimethyl-2-phenyl-2,4-dihydro-3H-pyrazol-3-one (**1g**) was synthesized with nearly quantitative yield (99%) according to the modified literature procedure [54]. However, the same procedure was less effective for the synthesis of diazide (**2g**) from **2**, where only 25% yield of **2g** was obtained. Thiocyanate (**1h**) was synthesized according to the previously developed procedure for the thiocyanation of the CH-acidic substrates [55]. Dimers (**1i** and **1j**) were synthesized by dehydrogenative dimerization of **1** employing mixed heterogeneous photocatalysis and homogeneous organocatalysis in photooxidative system N-hydroxyphthalimide (NHPI)/TiO$_2$ [56]. The reaction proceeded under air as the terminal oxidant to obtain the mixture of diastereomeric dimers with a total yield of 84%.

### 2.2. Study of the Fungicidal Activity of 4,4-Disubstituted Pyrazolones

In the next step we tested the fungicidal activity of the synthesized 4-disubstituted 2-phenyl-5-methylpyrazolin-3-ones against six species of phytopathogenic fungi characterized by high impact on crop production: *Venturia inaequalis*, *Rhizoctonia solani*, *Fusarium oxysporum*, *Fusarium moniliforme*, *Bipolaris sorokiniana*, and *Sclerotinia sclerotiorum*. Tests were performed by using the mycelium radial growth inhibition method in Petri dishes at a concentration of 10 mg/L and 5 mg/L in the culture medium. Kresoxim-methyl and triadimefon were used as the reference compounds (Table 1).

**Table 1.** In vitro fungicidal activity of the 4-substituted pyrazolin-3-ones *.

| | Compound | C, mg/L | Mycelium Growth Inhibition | | | | | |
| | | | *V. i.* | *R. s.* | *F. o.* | *F. m.* | *B. s.* | *S. s.* |
|---|---|---|---|---|---|---|---|---|
| 1 | **1a** (NO$_2$) | 10 | 100 | 100 | 61 | 100 | 100 | 88 |
| 2 | **1b** (OH) | 10 | 7 | 20 | 11 | 8 | 0 | 0 |
| 3 | **1c** (OCH$_3$) | 10 | 11 | 25 | 6 | 17 | 1 | 14 |
| 4 | **1d** (F) | 10 | 58 | 51 | 80 | 89 | 74 | 42 |
| 5 | **1e** (Cl) | 10 | 100 | 100 | 55 | 61 | 66 | 36 |

**Table 1.** *Cont.*

| | Compound | C, mg/L | Mycelium Growth Inhibition | | | | | |
|---|---|---|---|---|---|---|---|---|
| | | | *V. i.* | *R. s.* | *F. o.* | *F. m.* | *B. s.* | *S. s.* |
| 6 | **1f** | 10 | 100 | 100 | 31 | 83 | 60 | 54 |
| 7 | **2d** | 10 | 29 | 32 | 53 | 68 | 53 | 36 |
| 8 | **2e** | 10 | 100 | 100 | 85 | 100 | 77 | 100 |
| | | 5 | 100 | 100 | 68 | 100 | 52 | 31 |
| 9 | **2f** | 10 | 100 | 100 | 100 | 100 | 100 | 100 |
| | | 5 | 100 | 100 | 97 | 100 | 62 | 100 |
| 10 | **1g** | 10 | 33 | 38 | 8 | 22 | 6 | 1 |
| 11 | **2g** | 10 | 100 | 100 | 50 | 83 | 74 | 28 |
| 12 | **1h** | 10 | 22 | 39 | 11 | 11 | 16 | 11 |
| 13 | **1i** | 10 | 11 | 26 | 17 | 11 | 12 | 0 |
| 14 | **1j** | 10 | 11 | 24 | 6 | 19 | 14 | 0 |
| 15 | Kresoxim-methyl | 10 | 100 | 100 | 65 | 64 | 58 | 54 |
| 16 | Triadimefon | 10 | 49 | 58 | 62 | 82 | 64 | 63 |

\* Abbreviations used: V.i.—Venturia inaequalis, R.s.—Rhizoctonia solani, F.o.—Fusarium oxysporum, F.m.—Fusarium moniliforme, B.s.—Bipolaris sorokiniana, S.s.—Sclerotinia sclerotiorum. Fungicidal activity exceeding the standard (kresoxim-methyl) is highlighted.

The results of fungicidal tests showed that the nitro group in the fourth position of the pyrazolone ring is not the principal structural element responsible for the manifestation of fungicidal activity, as it was previously assumed [51,52]. The most effective derivatives were revealed: 4-methyl-4-chloropyrazolone (**1e**), 4-methyl-4-bromopyrazolone (**1f**),

4,4-dichloropyrazolone (**2e**), 4,4-dibromopyrazolone (**2f**) and 4,4-diazidopyrazolone (**2g**). Fluorinated derivatives (**1d** and **2d**) were moderately active. Monoazide derivative (**1g**) and thiocyanate (**1h**) showed low activity. The least active are pyrazolones with hydroxy and methoxy substituents (**1b** and **1c**), as well as dimeric compounds (**1i** and **1j**).

It should be noted that the synthesized C4-disubstituted pyrazolones have a different spectrum of activity than kresoxim-methyl. Pyrazolones **1a**, **1d**, **1f**, **2e**, **2f** and **2g** are more active against F. moniliforme, and 4,4-dibromopyrazolone **2f** also showed outstanding activity against S. sclerotiorum, the least affected by kresoxim-methyl, even at concentrations as low as 5 mg/L. Pyrazolones **1a**, **1d**, **1e**, **2e**, **2f** and **2g** showed activity at or above the kresoxim-methyl level against B. sorokiniana, with nitropyrazolone **1a** being the most active on the list. Pyrazolones **1a**, **1e**, **1f**, **2e**, **2f** and **2g** have excellent activity against V. inaequalis and R. solani: up to 100% mycelium growth inhibition at a concentration of 10 mg/L or 5 mg/L in the case of dihalogen pyrazolones **2e** and **2f**. Thus, the newly discovered fungicidal compounds can serve as a useful addition to the current range of agricultural fungicides for the control of fungi that are the least susceptible to existing fungicides.

Most of the previously known biologically active C4-disubstituted pyrazolones, with the exception of TELIN [43], belong to the class of spiro compounds bearing two C—C bonds at C4. Thus, the main structural feature that distinguishes the pyrazolones tested by us from other biologically active C4-disubstituted pyrazolones is the presence of a C4-heteroatom bond, which may be a key aspect for the manifestation of fungicidal activity.

## 3. Materials and Methods

K$_2$CO$_3$ (98%, extra pure, anhydrous, Thermo Scientific, Waltham, MA, USA), CH$_3$I (99.5%, Acros Organics, Geel, Belgium), Selectfluor (95%, Acros Organics), N-chlorosuccinimide (98%, Acros Organics), N-bromosuccinimide (99%, Acros Organics), NaN$_3$ (99.5%, Acros Organics), I$_2$ (99.8%, Acros Organics), NH$_4$SCN (97.5%, Acros Organics), Ce(NH$_4$)$_2$(NO$_2$)$_6$ (CAN, 98.5%, Thermo Scientific), N-hydroxyphthalimide (NHPI, 98%, Acros Organics) were used as is. Hombikat UV 100 (anatase, specific surface area, BET: 300 m$^2 \cdot$g$^{-1}$, primary crystal size according to Scherrer < 10 nm) was used as is. 4,5-dimethyl-2-phenyl-1,2-dihydro-3H-pyrazol-3-one **1** [51] and 4,5-dimethyl-4-nitro-2-phenyl-2,4-dihydro-3H-pyrazol-3-one **1a** [51] were synthesized according to the literature procedures. CH$_3$CN was distilled over P$_2$O$_5$, acetone was distilled over KMnO$_4$. 1,4-dioxane, DMSO, and glacial AcOH were used as is from commercial sources.

In all experiments, see Supplementary File, RT stands for 22–25 °C. $^1$H and $^{13}$C NMR spectra were recorded on a Bruker AVANCE II 300 and Bruker Fourier 300HD (300.13 and 75.47 MHz, respectively) spectrometers in CDCl$_3$ and DMSO-D$_6$. FT-IR spectra were recorded on Bruker Alpha instrument. High-resolution mass spectra (HR-MS) were measured on a Bruker maXis instrument using electrospray ionization (ESI). The measurements were performed in a positive ion mode (interface capillary voltage—4500 V); mass range from m/z 50 to m/z 3000 Da; external calibration with Electrospray Calibrant Solution (Fluka). A syringe injection was used for all acetonitrile solutions (flow rate 3 μL/min). Nitrogen was applied as a dry gas; the interface temperature was set at 180 °C.

For investigation of fungicidal activity, aseptic polystyrene Petri dishes (90 × 17 mm) were used. All glassware used for addition and mixing of acetone solutions of the tested compounds with agar medium were sterilized before usage. Experiments were performed in a laminar flow cabinet.

### 3.1. 4-Hydroxy-4,5-dimethyl-2-phenyl-2,4-dihydro-3H-pyrazol-3-one **1b** [57]

4-Hydroxy-4,5-dimethyl-2-phenyl-2,4-dihydro-3H-pyrazol-3-one **1b** was synthesized according to the literature procedure [53]. K$_2$CO$_3$ (20 mol.%, 0.4 mmol, 55 mg) was added to a solution of 4,5-dimethyl-2-phenyl-1,2-dihydro-3H-pyrazol-3-one **1** (2 mmol, 376 mg) in dioxane (30 mL). Then the flask was evacuated and filled with oxygen three times. The reaction mixture was stirred at 35 °C; for 24 h. Then the solution was evaporated, and the residue was purified by column chromatography on silica gel using the eluent

Petroleum Ether/EtOAc = 3/1 to afford **1b** as white crystals (377 mg, 1.85 mmol, 93%). Mp = 111–112 °C (lit. Mp = 113 °C [57]). **$^1$H NMR** (300.13 MHz, CDCl$_3$): δ = 7.84 (m, 2H, ArH), 7.38 (m, 2H, ArH), 7.18 (m, 1H, ArH), 4.58 (bs, 1H, OH), 2.18 (s, 3H), 1.53 (s, 3H). **$^{13}$C NMR** (75.47 MHz, CDCl$_3$): δ = 174.8, 163.4, 133.7, 129.0, 125.5, 119.0, 77.4, 22.3, 12.7.

### 3.2. 4-Methoxy-4,5-dimethyl-2-phenyl-2,4-dihydro-3H-pyrazol-3-one **1c**

CH$_3$I (1.5 mmol, 213 mg) was added to a stirred solution of 4-hydroxy-4,5-dimethyl-2-phenyl-2,4-dihydro-3H-pyrazol-3-one **1** (1 mmol, 204 mg) and K$_2$CO$_3$ (1.5 mmol, 207 mg) in acetone (10 mL). The reaction mixture was refluxed for 15 h, and the product formation was monitored by TLC (eluent Petroleum Ether/EtOAc = 5/1, R$_f$ = 0.5). Then the reaction mixture was diluted with CH$_2$Cl$_2$ (20 mL) and water (20 mL), the CH$_2$Cl$_2$ layer was separated, and the water layer was additionally extracted with CH$_2$Cl$_2$ (2 × 10 mL). Then, all organic extracts were combined, washed with water (2 × 20 mL), and dried over MgSO$_4$. The solvent was rotary evaporated and the residue was purified by column chromatography on silica gel using the eluent Petroleum Ether/EtOAc = 5/1 to afford **1c** as a slightly yellow liquid (23 mg, 0.11 mmol, 11%). **$^1$H NMR** (300.13 MHz, CDCl$_3$): δ = 7.92 (m, 2H), 7.41 (m, 2H), 7.20 (m, 1H), 3.20 (s, 3H), 2.15 (s, 3H), 1.47 (s, 3H). **$^{13}$C NMR** (75.47 MHz, CDCl$_3$): δ = 171.6, 161.3, 137.9, 129.0, 125.4, 118.6, 82.8, 54.3, 20.7, 13.1. **FT-IR** (thin layer): ν$_{max}$ = 1596, 1500, 1398, 1364, 1304, 1241, 1136, 1058, 759, 693 cm$^{-1}$. **HR-MS** (ESI): m/z = 236.1397, calcd. for C$_{12}$H$_{14}$N$_2$O$_2$+NH$_4$$^+$: 236.1394

### 3.3. 4-Fluoro-4,5-dimethyl-2-phenyl-2,4-dihydro-3H-pyrazol-3-one **1d**

Selectfluor$^{TM}$ (177 mg, 0.5 mmol) was added to a solution of 4,5-dimethyl-2-phenyl-1,2-dihydro-3H-pyrazol-3-one **1** (94 mg, 0.5 mmol) in CH$_3$CN (5 mL) at 60 °C. The reaction mixture was stirred for 1.5 h at 60 °C, then diluted with water (20 mL) and CH$_2$Cl$_2$ (10 mL). The CH$_2$Cl$_2$ layer was separated and the water layer was additionally extracted with CH$_2$Cl$_2$ (2 × 10 mL). All organic extracts were combined, washed with brine (20 mL), and dried over MgSO$_4$. Then the solvent was rotary evaporated, and the residue was purified by column chromatography on silica gel using the CH$_2$Cl$_2$ eluent to afford **1d** as yellow liquid (92 mg, 89%). **$^1$H NMR** (300.13 MHz, CDCl$_3$): δ = 7.94–7.80 (m, 2H), 7.46–7.34 (m, 2H), 7.25–7.14 (m, 1H), 2.19 (d, J = 1.6 Hz, 2H), 1.65 (d, J = 23.3 Hz, 2H). **$^{13}$C NMR** (75.47 MHz, CDCl$_3$): δ = 168.2 (d, J = 21.5 Hz), 158.1 (d, J = 16.6 Hz), 137.4, 129.0, 125.5, 118.5, 92.1 (d, J = 191.0 Hz), 18.8 (d, J = 27.0 Hz), 12.66 (d, J = 1.2 Hz). **$^{19}$F NMR** (282.47 MHz, CDCl$_3$): δ = −166.72 (q, J = 23.1 Hz). **FT-IR** (thin layer): ν$_{max}$ = 1736, 1598, 1501, 1368, 1144, 1117, 758. **HR-MS** (ESI): m/z = 229.0744, calcd. for C$_{11}$H$_{11}$FN$_2$O+Na$^+$: 229.0748

### 3.4. 4-Chloro-4,5-dimethyl-2-phenyl-2,4-dihydro-3H-pyrazol-3-one **1e** [58]

The solution of 4,5-dimethyl-2-phenyl-1,2-dihydro-3H-pyrazol-3-one **1** (1 mmol, 188 mg) in CH$_3$CN (20 mL) was added to a solution of N-chlorosuccinimide (NCS, 1.1 mmol, 147 mg) in CH$_3$CN (5 mL) dropwise in 5 min at room temperature. The reaction mixture was stirred for another 30 min, then the solvent was rotary evaporated, and the residue was purified by column chromatography on silica gel using the eluent Petroleum Ether/EtOAc = 5/1 to afford **1e** as white crystals (218 mg, 0.98 mmol, 98%). Mp = 68–69 °C (lit. Mp = 68 °C [58]). **$^1$H NMR** (300.13 MHz, CDCl$_3$): δ = 7.89 (m, 2H), 7.41 (m, 2H), 7.21 (m, 1H), 2.24 (s, 3H), 1.77 (s, 3H). **$^{13}$C NMR** (75.47 MHz, CDCl$_3$): δ = 169.8, 159.5, 137.6, 129.1, 125.6, 118.9, 62.7, 22.5, 12.9.

### 3.5. 4-Bromo-4,5-dimethyl-2-phenyl-2,4-dihydro-3H-pyrazol-3-one **1f** [58]

The solution of 4,5-dimethyl-2-phenyl-1,2-dihydro-3H-pyrazol-3-one **1** (1 mmol, 188 mg) in CH$_3$CN (20 mL) was added to a solution of N-bromosuccinimide (NBS, 1.1 mmol, 196 mg) in CH$_3$CN (5 mL) dropwise in 5 min at room temperature. The reaction mixture was stirred for another 30 min, then the solvent was rotary evaporated, and the residue was purified by column chromatography on silica gel using the eluent Petroleum Ether/EtOAc = 10/1 to afford **1f** (259 mg, 0.97 mmol, 97%) as yellow crystals. Mp = 81–82 °C (lit. Mp = 83 °C [58]). **$^1$H NMR** (300 MHz, Chloroform-d) δ 7.93–7.86 (m, 2H), 7.45–7.37 (m, 2H), 7.25–7.17 (m, 1H), 2.29

(s, 3H), 1.87 (s, 3H). **$^{13}$C NMR** (75.47 MHz, CDCl$_3$): δ = 170.3, 159.3, 137.7, 129.1, 125.6, 118.8, 52.7, 22.5, 13.2. **FT-IR** (thin layer): ν$_{max}$ = 1716, 1621, 1594, 1442, 1397, 1366, 1304, 1129, 767, 696, 635, 574, 511 cm$^{-1}$. HR-MS (ESI): m/z = 268.0515, calcd. for C$_{11}$H$_{11}$BrN$_2$O + H: 268.0506.

### 3.6. 4,4-Difluoro-5-methyl-2-phenyl-2,4-dihydro-3H-pyrazol-3-one **2d** [59]

Selectfluor (354 mg, 1 mmol) was added to a solution of 5-methyl-2-phenyl-1,2-dihydro-3H-pyrazol-3-one **2** (0.5 mmol, 87 mg) in CH$_3$CN (5 mL) under heating (60 °C). The reaction mixture was stirred at 60 °C for 30 min. Then the reaction mixture was diluted with water (20 mL) and CH$_2$Cl$_2$ (10 mL). The CH$_2$Cl$_2$ layer was separated and the water layer was additionally extracted with CH$_2$Cl$_2$ (2 × 10 mL). All organic extracts were combined, washed with brine (20 mL), and dried over MgSO$_4$. Then the solvent was rotary evaporated to afford **2d** as a yellow liquid (93 mg, 88%). **$^1$H NMR** (300.13 MHz, CDCl$_3$): δ = 7.87–7.68 (m, 2H), 7.44–7.30 (m, 2H), 7.25–7.15 (m, 1H), 2.23 (s, 3H). **$^{13}$C NMR** (75.47 MHz, CDCl$_3$): δ = 159.2 (t, J = 29.9 Hz), 152.0 (t, J = 23.0 Hz), 136.8, 129.2, 126.3, 118.6, 108.2 (t, J = 256.6 Hz), 11.8. **$^{19}$F NMR** (282.39 MHz, CDCl$_3$): δ = -123.1. **FT-IR** (thin layer): ν$_{max}$ = 1751, 1597, 1501, 1255, 1148, 1114, 756. **HR-MS** (ESI): m/z = 211.0684, calcd. for C$_{10}$H$_8$F$_2$N$_2$O + H$^+$: 211.0677.

### 3.7. 4,4-Dichloro-5-methyl-2-phenyl-2,4-dihydro-3H-pyrazol-3-one **2e** [60]

N-chlorosuccinimide (NCS, 2 mmol, 267 mg) was added in portions to a solution of 5-methyl-2-phenyl-1,2-dihydro-3H-pyrazol-3-one **2** (1 mmol, 174 mg) in CH$_3$CN (10 mL). The reaction mixture was stirred at room temperature for 30 min, then the solvent was rotary evaporated, and the residue was purified by column chromatography on silica gel using the CH$_2$Cl$_2$ eluent to afford **2e** as yellow crystals (185 mg, 0.76 mmol, 76%). Mp = 61–62 °C (lit. Mp = 61.5–63 °C [60]). **$^1$H NMR** (300.13 MHz, CDCl$_3$): δ = 7.93–7.81 (m, 2H), 7.52–7.41 (m, 2H), 7.33–7.20 (m, 1H), 2.39 (s, 3H). **$^{13}$C NMR** (75.47 MHz, CDCl$_3$): δ = 164.1, 155.6, 137.0, 129.2, 126.3, 119.0, 73.5, 12.2. **FT-IR** (thin layer): ν$_{max}$ = cm$^{-1}$. **HR-MS** (ESI): m/z = 243.0085, calcd. for C$_{10}$H$_{11}$Cl$_2$N$_2$O + H$^+$: 243.0086.

### 3.8. 4,4-Dibromo-5-methyl-2-phenyl-2,4-dihydro-3H-pyrazol-3-one **2f** [61]

N-bromosuccinimide (NBS, 2 mmol, 356 mg) was added in portions to a solution of 5-methyl-2-phenyl-1,2-dihydro-3H-pyrazol-3-one **2** (1 mmol, 174 mg) in CH$_3$CN (10 mL). The reaction mixture was stirred at room temperature for 30 min, then the solvent was rotary evaporated, and the residue was purified by column chromatography on silica gel using eluent Petroleum Ether/EtOAc = 10/1 to afford **2f** as yellow crystals (297 mg, 0.89 mmol, 89%). Mp = 76–78 °C (lit. Mp = 80–82 °C [61]). **$^1$H NMR** (300.13 MHz, CDCl$_3$): δ = 7.93–7.82 (m, 2H), 7.49–7.36 (m, 2H), 7.33–7.19 (m, 1H), 2.44 (s, 3H). **$^{13}$C NMR** (75.47 MHz, CDCl$_3$): δ = 165.3, 156.1, 137.1, 129.2, 126.2, 119.0, 46.2, 13.3. **FT-IR** (thin layer): ν$_{max}$ = 1722, 1592, 1491, 1364, 1283, 948, 804, 765, 694, 659, 632, 510 cm$^{-1}$. **HR-MS** (ESI): m/z = 330.9068, calcd. for C$_{10}$H$_8$Br$_2$N$_2$O + H: 330.9076.

### 3.9. 4-Azido-4,5-dimethyl-2-phenyl-2,4-dihydro-3H-pyrazol-3-one **1g**

4-azido-4,5-dimethyl-2-phenyl-2,4-dihydro-3H-pyrazol-3-one **1g** was synthesized according to the modified literature procedure [54]. I$_2$ (1.1 mmol, 279 mg) was added to a suspension of 4,5-dimethyl-2-phenyl-1,2-dihydro-3H-pyrazol-3-one **1** (1 mmol, 188 mg) and NaN$_3$ (3 mmol, 195 mg) in DMSO (10 mL). The reaction mixture was stirred at room temperature, and the formation of the product was monitored by TLC (eluent Petroleum Ether/EtOAc = 5/1, R$_f$ = 0.6). Then the reaction mixture was diluted with Na$_2$S$_2$O$_3$ solution (0.1 M, 30 mL) and CH$_2$Cl$_2$ (10 mL). The CH$_2$Cl$_2$ layer was separated and the water layer was additionally extracted with CH$_2$Cl$_2$ (2 × 10 mL). All organic extracts were combined, washed with water (2 × 20 mL), and dried over MgSO$_4$. Then the solvent was rotary evaporated, and the residue was purified by column chromatography on silica gel using the eluent Petroleum Ether/EtOAc = 5/1 to afford **1g** as a colorless liquid (226 mg, 0.99 mmol, 99%). **$^1$H NMR** (300.13 MHz, CDCl$_3$): δ = 7.88 (m, 2H), 7.41 (m, 2H), 7.21 (m, 1H), 2.14 (s,

3H), 1.63 (s, 3H). **¹³C NMR** (75.47 MHz, CDCl₃): δ = 170.7, 159.6, 137.6, 129.1, 125.6, 118.8, 66.0, 18.7, 13.2. **FT-IR** (thin layer): $\nu_{max}$ = 2109, 1719, 1597, 1501, 1399, 1366, 1312, 1246, 1137, 756, 692 cm⁻¹. **HR-MS** (ESI): m/z = 252.0860, calcd. for $C_{11}H_{11}N_5O+Na^+$: 252.0856.

### 3.10. 4,4-Diazido-5-methyl-2-phenyl-2,4-dihydro-3H-pyrazol-3-one **2g** *[54]*

4,4-diazido-5-methyl-2-phenyl-2,4-dihydro-3H-pyrazol-3-one **2g** was synthesized according to the literature procedure [54]. I₂ (2.2 mmol, 558 mg) was added to a suspension of 5-methyl-2-phenyl-1,2-dihydro-3H-pyrazol-3-one **2** (1 mmol, 174 mg) and NaN₃ (6 mmol, 390 mg) in DMSO (10 mL). The reaction mixture was stirred at room temperature, and the formation of the product was monitored by TLC (eluent Petroleum Ether/EtOAc = 20/1, $R_f$ = 0.5). Then the reaction mixture was diluted with Na₂S₂O₃ solution (0.1 M, 30 mL) and CH₂Cl₂ (10 mL). The CH₂Cl₂ layer was separated and the water layer was additionally extracted with CH₂Cl₂ (2 × 10 mL). All organic extracts were combined, washed with water (2 × 20 mL), and dried over MgSO₄. Then the solvent was rotary evaporated, and the residue was purified by column chromatography on silica gel using the eluent Petroleum Ether/EtOAc = 20/1 to afford **2g** as brown liquid (62 mg, 0.25 mmol, 25%). **¹H NMR** (300 MHz, Chloroform-d) δ 7.96–7.85 (m, 2H), 7.53–7.39 (m, 2H), 7.34–7.21 (m, 1H), 2.18 (s, 3H). **¹³C NMR** (75.47 MHz, CDCl₃): δ = 163.9, 155.6, 136.8, 129.2, 126.2, 118.8, 13.0.

### 3.11. 4,5-Dimethyl-2-phenyl-4-thiocyanato-2,4-dihydro-3H-pyrazol-3-one **1h** *[55]*

4,5-dimethyl-2-phenyl-4-thiocyanato-2,4-dihydro-3H-pyrazol-3-one **1h** was synthesized according to the literature procedure [55]. Ce(NH₄)₂(NO₂)₆ (CAN, 3 mmol, 1644 mg) was added in portions to a solution of 4,5-dimethyl-2-phenyl-1,2-dihydro-3H-pyrazol-3-one **1** (1 mmol, 188 mg) and NH₄SCN (3 mmol, 228 mg) in AcOH (10 mL) within 30 min. The reaction mixture was stirred at room temperature, and the formation of the product was monitored by TLC (eluent Petroleum Ether/EtOAc = 5/1). Then the reaction mixture was diluted with CH₂Cl₂ (20 mL) and water (20 mL). The CH₂Cl₂ layer was separated and the water layer was additionally extracted with CH₂Cl₂ (2 × 10 mL). All organic extracts were combined, washed with water (2 × 20 mL), and dried over MgSO₄. Then the solvent was rotary evaporated, and the residue was purified by column chromatography on silica gel using the eluent Petroleum Ether/EtOAc = 5/1 to afford **1h** as slightly yellow crystals (97 mg, 0.40 mmol, 40%). Mp = 99–100 °C (lit. Mp = 97–99 °C [55]). **¹H NMR** (300.13 MHz, CDCl₃): δ = 7.92 (m, 2H), 7.48 (m, 2H), 7.29 (m 1H), 2.35 (s, 3H), 1.76 (s, 3H). **¹³C NMR** (75.47 MHz, CDCl₃): δ = 169.4, 157.6, 137.1, 129.1, 126.1, 119.3, 107.2, 56.8, 18.7, 13.4. **FT-IR** (thin layer): $\nu_{max}$ = cm⁻¹. **HR-MS** (ESI): m/z = 268.0506, calcd. for $C_{12}H_{11}N_3OS+Na^+$: 268.0515.

### 3.12.
### 4,4′,5,5′-Tetramethyl-2,2′-diphenyl-2,2′,4,4′-tetrahydro-3H,3′H-[4,4′-bipyrazole]-3,3′-dione, meso **1i** and racemic **1j** *[62]*

Dimers were synthesized according to the modified literature procedure [56]. To a 50 mL round-bottomed flask 4,5-dimethyl-2-phenyl-1,2-dihydro-3H-pyrazol-3-one **1** (1 mmol, 188 mg), TiO₂ Hombikat UV 100 (20 mg), N-hydroxyphthalimide (20 mol%, 0.2 mmol, 32.6 mg) and a solvent (CH₃CN, 2 mL) were placed. The resulting suspension was sonicated for 1 min. The stirred reaction mixture was irradiated with 10 W Blue LED (443 nm) at room temperature until full conversion of **1** (2 h, monitored by TLC, eluent CH₂Cl₂). Upon completion, the reaction mixture was diluted with CH₂Cl₂ (20 mL) and water (20 mL). The CH₂Cl₂ layer was separated and the water layer was additionally extracted with CH₂Cl₂ (2 × 10 mL). All organic extracts were combined, washed with NaHCO₃ saturated solution (20 mL) and water (20 mL), and dried over MgSO₄. Then the solvent was rotary evaporated, and the residue was purified by column chromatography on silica gel using the eluent CH₂Cl₂/EtOAc = 40/1 to afford diastereomeric dimers **1i** (54%, 0.27 mmol, 100 mg) and **1j** (30%, 0.15 mmol, 56 mg) as white crystals.

**Meso-3,3′,4,4′-tetramethyl-1,1′-diphenyl-[4,4′-bipyrazol]-5,5′-dione, 1i** Mp = 161–162 °C (Lit. Mp = 163–164 °C [62]); **¹H NMR** (300.13 MHz, CDCl₃): δ = 7.93–7.87 (m, 2H), 7.46–7.38

(m, J = 10.8, 5.1 Hz, 2H), 7.22 (t, J = 7.4 Hz, 1H), 1.93 (s, 6H), 1.73 (s, 6H); **¹³C NMR** (75.47 MHz, CDCl₃): δ = 173.1, 161.9, 137.6, 129.1, 125.7, 119.0, 54.4, 14.7, 14.6.

**Racemic-3,3′,4,4′-tetramethyl-1,1′-diphenyl-[4,4′-bipyrazol]-5,5′-dione, 1j** Mp = 141–142 °C (Lit. Mp = 140–141 °C [62]); **¹H NMR** (300.13 MHz, CDCl₃): δ = 7.85 (d, J = 7.9 Hz, 4H), 7.45–7.31 (m, 4H), 7.18 (t, J = 7.3 Hz, 2H), 2.19 (s, 6H), 1.60 (s, 6H); **¹³C NMR** (75.47 MHz, CDCl₃): δ = 173.1, 159.8, 137.7, 129.0, 125.4, 119.3, 55.7, 16.0, 15.4.

### *3.13. Investigation of Fungicidal Activity (Table 1)*

The fungicidal activity was investigated according to a standard procedure [63–69]. The strains for fungicidal studies were obtained from the working collection of the All-Russian Research Institute for Phytopathology (B. Vyazemy, Moscow reg., Russia): Venturia inaequalis (V.i.) MRA-16-2, Rhizoctonia solani (R.s.) 100063, Fusarium oxysporum (F.o.) FO-8, Fusarium moniliforme (F.m.) 100146, Bipolaris sorokiniana (B.s.) MRB(V)-1, Sclerotinia sclerotiorum (S.s.) 100033. The test substances preliminarily dissolved in acetone (concentration 1 or 0.5 mg mL⁻¹) were introduced into liquid potato sugar agar having a temperature of 50–55 °C so that the final concentration of the substance in the nutrient medium was 10 mg L⁻¹ or 5 mg L⁻¹ (0.9 mL of solution in acetone per 90 mL of agar). After mixing, the agar was poured into sterile Petri dishes and cooled to room temperature. Pieces of mycelium from the peripheral growth zone of mycelium culture incubated for 3–5 days were transferred to Petri dishes with diluted tested compounds using a needle. The control was a colony grown in the same nutrient medium without the addition of the active substance (acetone without substance was added). After inoculation for 72 h, the diameters of the formed fungal colonies were measured. The indicator of fungicidal activity was the suppression of mycelium growth in comparison with the control, calculated as $[(D_c - D_s)/D_c] \times 100\,\%$, in which $D_c$ is the diameter of the colony of fungus in the control medium and $D_s$ is the diameter of the colony in the medium with the test substance added.

### 4. Conclusions

Pronounced broad spectrum fungicidal activity was found to be characteristic for a wide range of C4-disubstituted pyrazolin-3-ones. This discovery shows that 4-nitropyrazolin-3-ones, previously reported as a novel class of fungicides, are actually the subgroup of a more diverse type of fungicidal structures, which can be explored further. Currently, the most active compounds are 4,4-dichloro-, 4,4-dibromo pyrazolin-3-ones **2e** and **2f**. 4-Methyl-4-chloro-, 4-methyl-4-bromo- and 4,4-diazidopyrazolones showed somewhat lower activity. For the most active structures, efficient synthesis procedures have been proposed that allow for obtaining substances with high to quantitative yields. The ease of synthesis, the availability of reagents, and high fungicidal activity comparable to that of commercial fungicides, make the discovered 4-disubstituted pyrazolin-3-ones attractive candidates for the role of a new class of fungicidal compounds.

**Supplementary Materials:** The following supporting information can be downloaded at: https://www.mdpi.com/article/10.3390/agrochemicals2010004/s1, ¹H and ¹³C NMR spectra of the synthesized compounds, FT-IR and HRMS data for new compounds.

**Author Contributions:** Conceptualization, I.B.K. and A.O.T.; investigation, E.R.L., A.S.B., A.L.A. and I.B.K.; writing—original draft preparation, E.R.L.; writing—review and editing, I.B.K., A.S.B. and A.O.T.; supervision, A.I.I., A.P.G. and A.O.T.; project administration, A.I.I. and A.P.G. All authors have read and agreed to the published version of the manuscript.

**Funding:** This research was funded by Russian Science Foundation, grant number 19-73-20190.

**Institutional Review Board Statement:** Not applicable.

**Informed Consent Statement:** Not applicable.

**Data Availability Statement:** Not applicable.

**Conflicts of Interest:** The authors declare no conflict of interest.

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
