# Peer review of "4-Disubstituted Pyrazolin-3-Ones—Novel Class of Fungicides against Phytopathogenic Fungi"

_agrochemicals, doi:10.3390/agrochemicals2010004_

Round 1

Reviewer 1 Report

Dear Authors,

is a very interesting manuscript.

I have a few comments on the methodology of testing the impact of individual substituted pyrazolin-3-ones. In my opinion, as a mycologist, the method of determining the inhibitory effect of individual compounds is archaic. It should at most be a supplement to the basic method, i.e. microdilutions on titration plates. In terms of the visual effect of individual substances, it looks good, but does it really indicate the real effect of the antifungal substance on the tested fungus strain?

We tried to use an identical method to study the effect of individual antifungal drugs on strains of different dermatophytes. But the results obtained with the mycelium radial growth inhibition method in Petri dishes did not compare well with the MIC value obtained with the M38 CLSI standard and the correlation with in vivo treatment.

The first remark concerns the solvent used to prepare the dilutions. Why acetone? acetone is toxic to fungi. To eliminate the effect of the solvent, DMSO and/or water are used as standard, depending on the substance. In the case of adding substances to the agar medium, there is always a risk of uneven distribution of the substance in the medium and whether there is a non-specific binding of the substance with the components of the medium and the solvent.

We tried to use an identical method to study the effect of individual antifungal drugs on strains of different dermatophytes. But the results obtained with the mycelium radial growth inhibition method in Petri dishes did not compare well with the MIC value obtained with the M38 CLSI standard and the correlation with in vivo treatment.

The first remark concerns the solvent used to prepare the dilutions. Why acetone? acetone is toxic to fungi. To eliminate the effect of the solvent, DMSO and/or water are used as standard, depending on the substance. In the case of adding substances to the agar medium, there is always a risk of uneven distribution of the substance in the medium and whether there is a non-specific binding of the substance with the components of the medium and the solvent.

Another issue is that the results are different when we cover the plate with a standardized suspension of the tested strain, and it is different to cut out a hole in the center of the plate with a corkscrew and the implementation of ready-made mycelium from the substrate without the substance. Needless to say, there will be no diffusion of substances from the new substrate to the old substrate. Another issue, since even the tested strain does not grow on a new plate, does this mean that over time the mechanisms of resistance and adaptation of the test substance will be activated?

Unfortunately, such a methodology would not pass in any typical mycological journal. it is different, for example, to saturate the disk with the tested concentration of the substance, cover the plate with the changed suspension of the tested strain and observe the size of the growth inhibition zone.

Unfortunately, the basis is to perform microdilutions/macrodilutions in a liquid medium, because only this allows to demonstrate a fairly real flow of the substance on the tested strain by the fact that the strain is immersed in the test substance from the very beginning.

Author Response

Dear Reviewer,

Thank you for the careful evaluation of our manuscript and your comments. The manuscript was corrected accordingly, the track-changes version of the manuscript file is attached to the submission. The step-by-step answers to the comments of Reviewer 1 are listed below. Answers to all Reviewers are attached as pdf file.

Reviewer’s comment: Dear Authors,

is a very interesting manuscript.

I have a few comments on the methodology of testing the impact of individual substituted pyrazolin-3-ones. In my opinion, as a mycologist, the method of determining the inhibitory effect of individual compounds is archaic. It should at most be a supplement to the basic method, i.e. microdilutions on titration plates. In terms of the visual effect of individual substances, it looks good, but does it really indicate the real effect of the antifungal substance on the tested fungus strain?

Answer: The radial mycelium growth inhibition method on agar medium containing fungicide may be more labour- and material- consuming, however, it is still widely used, especially for testing of plant protection fungicides, as a standard method of choice. Some examples of works from different research groups using this approach are listed below. The estimation of real effect of the antifungal substance on the tested fungus strain by this method is correct and reproducible when one compares the effect of the tested compound with the effect of a reference compound and results obtained by the same method. The wide usage of this method in the field of fungicide development for plant protection is one of the reasons for its convenience: it makes results from different researchers easily comparable.

  1. Agric. Food Chem. 2021, 69, 8358 https://doi.org/10.1021/acs.jafc.1c01189
  2. Agric. Food Chem. 2018, 66, 24, 6239 https://doi.org/10.1021/acs.jafc.8b02151
  3. Agric. Food Chem. 2021, 69, 40, 12048 https://doi.org/10.1021/acs.jafc.1c03325
  4. Agric. Food Chem.2012, 60, 23, 5813. https://doi.org/10.1021/jf300730f

Eur. J. Plant. Pathol. 2016, 144, 337. https://doi.org/10.1007/s10658-015-0771-z

Food Control, 2014, 41, 116. https://doi.org/10.1016/j.foodcont.2014.01.010

World J. Microbiol. Biotechnol. 2008, 24, 1445. https://doi.org/10.1007/s11274-007-9636-8

Tetrahedron, 2018, 74, 672. https://doi.org/10.1016/j.tet.2017.12.043

Eur. J. Med. Chem., 2017, 14, 141. https://doi.org/10.1016/j.ejmech.2017.09.009

Eur. J. Med. Chem., 2011, 46, 4374. https://doi.org/10.1016/j.ejmech.2011.07.008

Bioorg. Med. Chem., 2002, 10, 4029. https://doi.org/10.1016/S0968-0896(02)00302-4

Microbial Pathogenesis, 2016, 95, 186. https://doi.org/10.1016/j.micpath.2016.04.012

Eur. J. Med. Chem., 2011, 46, 364. https://doi.org/10.1016/j.ejmech.2010.10.022

Additional references were added to the experimental part.

Reviewer’s comment: We tried to use an identical method to study the effect of individual antifungal drugs on strains of different dermatophytes. But the results obtained with the mycelium radial growth inhibition method in Petri dishes did not compare well with the MIC value obtained with the M38 CLSI standard and the correlation with in vivo treatment.

Answer: The procedure we used is recommended for testing of compounds against phytopathogenic fungi and may be not suitable for dermatophytes. MIC values for different standard methods can differ seriously, that is why it is important to compare values obtained by the same methods and use reference compounds. We used two compounds of different fungicide classes: triadimefon (triazole fungicide, sterol biosynthesis inhibitors), and kresoxim-methyl (strobilurin fungicide, Quinone "outside" site of the bc1 complex inhibitor).

Reviewer’s comment: The first remark concerns the solvent used to prepare the dilutions. Why acetone? acetone is toxic to fungi. To eliminate the effect of the solvent, DMSO and/or water are used as standard, depending on the substance.

Answer: Both acetone or DMSO are used standardly in this method. The possible effect of solvent eliminated by addition of the same solvent to the control Petri dish (the same volume of solvent as in test dish but without dissolved compound). We have added this important note to the experimental part and thank the Reviewer for this comment:

Old version: The control was a colony grown in the same nutrient medium without the addition of the active substance.

Corrected version: The control was a colony grown in the same nutrient medium without the addition of the active substance (acetone without substance was added).

Reviewer’s comment: In the case of adding substances to the agar medium, there is always a risk of uneven distribution of the substance in the medium and whether there is a non-specific binding of the substance with the components of the medium and the solvent.

Answer: The solution of tested compound added to liquid potato sugar agar at 50–55 °C and mixed well to guarantee even distribution.

Reviewer’s comment: Another issue is that the results are different when we cover the plate with a standardized suspension of the tested strain, and it is different to cut out a hole in the center of the plate with a corkscrew and the implementation of ready-made mycelium from the substrate without the substance. Needless to say, there will be no diffusion of substances from the new substrate to the old substrate. Another issue, since even the tested strain does not grow on a new plate, does this mean that over time the mechanisms of resistance and adaptation of the test substance will be activated?

Answer: Thank you for this comment. Indeed, there are always many factors in such tests with living fungi. The key to correct interpretation of the results is procedure standardization and comparison between results obtained by the same procedure. For this reason, we have used two reference compounds, triadimefon and kresoxim-methyl, in our tests and used procedure employed in a significant number of papers on fungicide development for plant protection. In addition, in this paper we wanted to focus on the main insight – that that 4-mono and 4,4-dihalogenated pyrazolin-3-ones demonstrate fungicidal activity comparable to that of 4-nitropyrazolin-3-ones and other modern fungicides. This means that -NO2 group in 4-nitropyrazolin-3-ones, which was previously considered as obligatory for activity, can be replaced by other group of comparable size and electronic properties without the loss of fungicidal properties. This discovery significantly expands the scope of potent new fungicides based on pyrazolin-3-one fragment.

Reviewer’s comment: Unfortunately, such a methodology would not pass in any typical mycological journal. it is different, for example, to saturate the disk with the tested concentration of the substance, cover the plate with the changed suspension of the tested strain and observe the size of the growth inhibition zone.

Unfortunately, the basis is to perform microdilutions/macrodilutions in a liquid medium, because only this allows to demonstrate a fairly real flow of the substance on the tested strain by the fact that the strain is immersed in the test substance from the very beginning.

Answer: Thank you for reading and generally high evaluation of the manuscript, as well as for constructive criticism. As we noted above, we used one of the standard and widely used procedures of fungicide testing used in the field of crop protection. It should be noted that there is many other methods with their own advantages and disadvantages. The used method is most widely used for activity testing against molds [Balouiri, M.; Sadiki, M.; Ibnsouda, S. K. Methods for in Vitro Evaluating Antimicrobial Activity: A Review. Journal of Pharmaceutical Analysis 2016, 6 (2), 71–79. https://doi.org/10.1016/j.jpha.2015.11.005.], especially for plant pathogens [Ncama, K.; Mditshwa, A.; Tesfay, S. Z.; Mbili, N. C.; Magwaza, L. S. Topical Procedures Adopted in Testing and Application of Plant-Based Extracts as Bio-Fungicides in Controlling Postharvest Decay of Fresh Produce. Crop Protection 2019, 115, 142–151. https://doi.org/10.1016/j.cropro.2018.09.016.]. However, it is also used in clinical practice [Berkow, E. L.; Lockhart, S. R.; Ostrosky-Zeichner, L. Antifungal Susceptibility Testing: Current Approaches. Clin Microbiol Rev 2020, 33 (3), e00069-19. https://doi.org/10.1128/CMR.00069-19.]. Of course, we will have to perform many additional tests in the future research, including tests on living plants, but we sure that chosen methodology is well suited for the purposed of present pioneering report.

Reviewer 2 Report

This manuscript reports on the fungicidal activity of a series of new pyrazolinone derivatives. It is concise and well written, and I can only remark that in methods, probably due to inadvertence, authors did not specify that control plates received the same amount of acetone used for dissolving the tested products. Moreover, at line 112 it is said that assays were carried out at the concentrations of 10 and 5 mg/L; however, neither results concerning the latter concentration were reported, nor the latter concentration was considered in methods. If activity at 5 mg/L is not relevant, it should not be mentioned at all. Besides these adjustments, the paper can be accepted for publication in Agrochemicals after minor revision according to the below list of corrections.

Line 18: 'and' not in italics;

line 31: correct to 'most';

line 59: correct to 'disubstituted';

line 60 and Fig. 1: correct to 'antimicrobial';

line 62: correct to 'was not';

line 72 and throughout the manuscript: compound number should be in round brackets on first mention (1); then, numbers can be used without parentheses when referring to the named products;

lines 109-111: abbreviations of species names are only used in Table 1; hence, they should be provided as footnotes in that table rather than in the text;

line 112: correct to 'at a concentration of 10 mg/L';

line 113: 'Triadimefon' in lowercase initials; the same to be checked throughout the paper for names of other fungicides;

line 115: 'Vitro Fungicidal Activity' in lowercase initials;

line 116: correct to 'Fungicidal activity exceeding the standard (kresoxim-methyl) is highlighted';

line 127: from now on, use the abbreviated species names (e.g. F. moniliforme);

line 128: correct to '...against S. sclerotiorum, the...';

lines 156-158: correct to 'K2CO3 (20 mol.%, 0.4 mmol, 55 mg) was added to a solution of 1 (2 mmol, 376 mg) in dioxane (30 mL)'; likewise, the corresponding introductory sentences should be corrected in the descriptions of all products;

line 283: use 'min'; add 'and' before 'the formation';

line 343: delete 'one' and 'to' after 'high'.

Author Response

Dear Reviewer,

Thank you for the careful evaluation of our manuscript and your comments. The manuscript was corrected accordingly, the track-changes version of the manuscript file is attached to the submission. The step-by-step answers to the comments of Reviewer 2 are listed below. Answers to all Reviewers are attached as pdf file.

Reviewer’s comment: This manuscript reports on the fungicidal activity of a series of new pyrazolinone derivatives. It is concise and well written, and I can only remark that in methods, probably due to inadvertence, authors did not specify that control plates received the same amount of acetone used for dissolving the tested products.

Answer: We have added this important note to the experimental part and thank the Reviewer for this comment:

Old version: The control was a colony grown in the same nutrient medium without the addition of the active substance.

Corrected version: The control was a colony grown in the same nutrient medium without the addition of the active substance (acetone without substance was added).

Reviewer’s comment: Moreover, at line 112 it is said that assays were carried out at the concentrations of 10 and 5 mg/L; however, neither results concerning the latter concentration were reported, nor the latter concentration was considered in methods. If activity at 5 mg/L is not relevant, it should not be mentioned at all.

Answer: Testing in lower concentration (5 mg/L) were performed for most active compounds 2e and 2f showing complete fungal growth inhibition at standard concentration 10 mg/L (see entries 8-9 in Table 1). The head of Table 1 and the corresponding text in experimental part were corrected accordingly.

Reviewer’s comment: Besides these adjustments, the paper can be accepted for publication in Agrochemicals after minor revision according to the below list of corrections.

Line 18: 'and' not in italics;

line 31: correct to 'most';

line 59: correct to 'disubstituted';

line 60 and Fig. 1: correct to 'antimicrobial';

line 62: correct to 'was not';

line 72 and throughout the manuscript: compound number should be in round brackets on first mention (1); then, numbers can be used without parentheses when referring to the named products;

lines 109-111: abbreviations of species names are only used in Table 1; hence, they should be provided as footnotes in that table rather than in the text;

line 112: correct to 'at a concentration of 10 mg/L';

line 113: 'Triadimefon' in lowercase initials; the same to be checked throughout the paper for names of other fungicides;

line 115: 'Vitro Fungicidal Activity' in lowercase initials;

line 116: correct to 'Fungicidal activity exceeding the standard (kresoxim-methyl) is highlighted';

line 127: from now on, use the abbreviated species names (e.g. F. moniliforme);

line 128: correct to '...against S. sclerotiorum, the...';

lines 156-158: correct to 'K2CO3 (20 mol.%, 0.4 mmol, 55 mg) was added to a solution of 1 (2 mmol, 376 mg) in dioxane (30 mL)'; likewise, the corresponding introductory sentences should be corrected in the descriptions of all products;

line 283: use 'min'; add 'and' before 'the formation';

line 343: delete 'one' and 'to' after 'high'.

Answer: Thank you for careful reading, remarks and high evaluation of the manuscript. All proposed corrections were made.

Reviewer 3 Report

Manuscript presented by Elena R. Lopat’eva et al. shows a study about 4-disubstituted pyrazolin-3-ones as candidates of new fungicides. The topic is very important due to industry aspects.

An already well written and prepared manuscript. Easy to read and follow. Some aspects should be improved. I recommend the article to publish but first the paper should be corrected. My decision – reconsider after minor revision. Comments to be considered, in order to further improve the manuscript quality:

(1) Please define which compounds are new connections and which were synthesized previously. For new compound please add HR-MS and IR spectra to supporting information.

(2) In supporting information please add more specification like NMR inert standard, what technique was used for preparation of FTIR, type of column used in HR-MS and ect. Consider including this information in the main manuscript as well.

(3) To better clarity of the manuscript please part 2. Result and discussion divide into several section include synthesis part and biological results.

(4) In the section 3. Materials and Methods please add information about preparation of the fungicidal activity tests.

(5) Please add code for species of fungi used in presented study (last paragraph page 4).

(6) In the introduction, information about the importance of heterocycles in modern organic chemistry should be added. It should be underline the role of  pyrazolinones derivatives. Include the information about synthesis methods of this class of compounds is also welcome. These aspects were described recently in other MDPI publications and should be adding to manuscript: Molecules, 27, 8409 (2022) DOI: https://doi.org/10.3390/molecules27238409; Molecules, 26, 1364 (2021) DOI: https://doi.org/10.3390/molecules26051364;

(7) In order to unification symbols in manuscript, please “Me” group replace on “CH3”.

(8) Please specify PE/EA.

(9) Superscripts and subscripts as well as commas and periods should be checked (eg Rf). Avoid extra spaces and enters. Correct in whole manuscript.

(10) The English correction is necessary.

Author Response

Dear Reviewer,

Thank you for the careful evaluation of our manuscript and your comments. The manuscript was corrected accordingly, the track-changes version of the manuscript file is attached to the submission. The step-by-step answers to the comments of Reviewer 3 are listed below. Answers to all Reviewers are attached as pdf file.

Reviewer’s comment: Manuscript presented by Elena R. Lopat’eva et al. shows a study about 4-disubstituted pyrazolin-3-ones as candidates of new fungicides. The topic is very important due to industry aspects.

An already well written and prepared manuscript. Easy to read and follow. Some aspects should be improved. I recommend the article to publish but first the paper should be corrected. My decision – reconsider after minor revision.

Answer: Thank you for reading and high evaluation of the manuscript.

Reviewer’s comment: Comments to be considered, in order to further improve the manuscript quality:

(1) Please define which compounds are new connections and which were synthesized previously. For new compound please add HR-MS and IR spectra to supporting information.

Answer: References were added to all previously characterized compounds. HR-MS and IR spectra were added to supporting information for new compounds 1c, 1d, and 1g.

Reviewer’s comment: (2) In supporting information please add more specification like NMR inert standard, what technique was used for preparation of FTIR, type of column used in HR-MS and ect. Consider including this information in the main manuscript as well.

Answer: The missing information was added to supporting information. In NMR spectra, residual signals of CDCl3 (7.26 in 1H NMR, 77.16 in 13C NMR) or DMSO-d6 (2.50 in 1H NMR, 39.52 in 13C NMR) were used as reference signals for precise chemical shift determination. IR spectra were registered in KBr pellets for solid compounds, and liquid compounds were placed between two KBr windows to make a thin layer. For HR-MS the direct syringe injection in acetonitrile solutions (flow rate 3 µL/min).

Reviewer’s comment: (3) To better clarity of the manuscript please part 2. Result and discussion divide into several section include synthesis part and biological results.

Answer: The section was divided into synthetic part and biological results according to the Reviewer’s advice.

Reviewer’s comment: (4) In the section 3. Materials and Methods please add information about preparation of the fungicidal activity tests.

Answer: Information about the preparation of the fungicidal activity tests was added:

Added text: For investigation of fungicidal activity aseptic polystyrene Petri dishes (90 × 17 mm) were used. All glassware used for addition and mixing of acetone solutions of the tested compounds with agar medium was sterilized before usage. Experiments were performed in laminar flow cabinet.

Reviewer’s comment: (5) Please add code for species of fungi used in presented study (last paragraph page 4).

Answer: Codes of fungi strains used are given in the experimental part. We decided to not overload the discussion section with repetition of this information.

Reviewer’s comment: (6) In the introduction, information about the importance of heterocycles in modern organic chemistry should be added. It should be underline the role of  pyrazolinones derivatives. Include the information about synthesis methods of this class of compounds is also welcome. These aspects were described recently in other MDPI publications and should be adding to manuscript: Molecules, 27, 8409 (2022) DOI: https://doi.org/10.3390/molecules27238409; Molecules, 26, 1364 (2021) DOI: https://doi.org/10.3390/molecules26051364;

Answer: The sentence highlighting the importance of pyrazolone heterocycles was added to the introduction section with the citation of the proposed papers.

Reviewer’s comment:  (7) In order to unification symbols in manuscript, please “Me” group replace on “CH3”.

Answer: “Me” was replaced by “CH3”.

Reviewer’s comment: (8) Please specify PE/EA.

Answer: PE/EA was rewritten as “Petroleum Ether/EtOAc”.

Reviewer’s comment: (9) Superscripts and subscripts as well as commas and periods should be checked (eg Rf). Avoid extra spaces and enters. Correct in whole manuscript.

Answer: The corrections were made in whole manuscript.

Reviewer’s comment: (10) The English correction is necessary.

Answer: The language of the manuscript was polished.

Round 2

Reviewer 1 Report

none